# Numerical Modeling of Shockwave Treatment of Knee Joint

**DOI:** 10.3390/ma14247678

**Published:** 2021-12-13

**Authors:** Galina Eremina, Alexey Smolin

**Affiliations:** Institute of Strength Physics and Materials Science of Siberian Branch Russian Academy of Sciences, 2/4, Pr. Akademicheskii, 634055 Tomsk, Russia; asmolin@ispms.ru

**Keywords:** bone diseases and mechanical properties, poroelasticity, computer simulation, shock wave treatment

## Abstract

Arthritis is a degenerative disease that primarily affects the cartilage and meniscus of the knee joint. External acoustic stimulation is used to treat this disease. This article presents a numerical model of the knee joint aimed at the computer-aided study of the regenerative effects of shockwave treatment. The presented model was verified and validated. A numerical analysis of the conditions for the regeneration of the tissues of the knee joint under shockwave action was conducted. The results allow us to conclude that to obtain the conditions required for the regeneration of cartilage tissues and meniscus (compressive stresses above the threshold value of 0.15 MPa to start the process of chondrogenesis; distortional strains above the threshold value of 0.05% characterized by the beginning of the differentiation of the tissues in large volumes; fluid pressure corresponding to the optimal level of 68 kPa to transfer tissue cells in large volumes), the energy flux density of therapeutic shockwave loading should exceed 0.3 mJ/mm^2^.

## 1. Introduction

Degenerative diseases of the knee joint, such as arthritis and others, significantly reduce a person’s quality of life. The average age of the onset of degenerative diseases of this type is 45 years, which leads to disability among a significant part of the working population. However, arthritis can also result from injury [1]. In the end, osteoarthritis has a huge financial toll on the healthcare system [2]. The cartilage plates and meniscus have the function of redefining the loads and helping dissipate the energy obtained as a result of the dynamic effect on the joint. These structural elements of the knee joint are most susceptible to degenerative arthritic changes. That is why the problems of regeneration of cartilaginous plates and meniscus are given a lot of attention in modern medicine. Most acutely, this problem affects the elderly, in whom, due to age, degradation processes begin to appear in the soft tissues and bone tissues of the joints. Former athletes are also highly susceptible to degenerative changes. They have a high level of abrasion of cartilage tissue and soft tissue trauma (meniscus rupture). Surgery is performed in case of catastrophic degenerative changes in the elements of the lower limbs of the skeleton. Pharmacological therapy (based on taking medications, injection, etc.) [3] and non-pharmacological therapy (based on a mechanobiological concept that assumes the division and differentiation of biological cells under the influence of mechanical stimuli) are used to treat mild to moderate degenerative changes in the human musculoskeletal system [4]. Non-pharmacological therapy consists of a set of physical exercises [5] and external physical influence. Shockwave treatment is one of the methods of external exposure therapy [6].

For the treatment of degenerative diseases of the lower extremities, combined therapy is used based on medication (by hormonal and anti-inflammatory drugs) and mechanical treatment (massages, physical exercises, external mechanical stimulation devices). The effect of external mechanical stimulation is based on mechanobiological principles, the essence of which is that a certain level of pressure (mechanical stress) and deformations leads to the growth and differentiation of a certain type of biological tissue. Thus, the regeneration of cartilaginous tissue is facilitated by compressive stresses with an amplitude of 0.15 to 2 MPa (proliferation of chondrocytes) [7]. It was also noted that at stresses lower than 0.003 MPa, chondrogenesis and osteogenesis do not occur, and stresses of the order of compression of 0.7–0.8 MPa are most favorable for the formation of cartilaginous tissue. The optimum for the migration of living cells is the fluid pressure in the pores in the range from 20 kPa to 2 MPa (68 kPa is the most favorable value) [8]. One of the parameters of cartilage tissue differentiation is a value of distortional strain in the range from 0.05 to 1.1% [9,10].

The peculiarities of the physiological state (age, illness, injury) of the patient require an individual selection of the parameters of the mechanical treatment. The energy flux density (EFD) is the main characteristic of the shock wave process for biomedical purposes [11]. The loading is performed through metal or ceramic plates (applicators) of various shapes with a certain level of energy flux density, which is calculated from the parameters of the plates. Currently, active research is underway to determine the optimal loading parameters that contribute to the regeneration of bone and cartilage tissue. Shockwave therapy has been shown to be highly effective in the treatment of knee osteoarthritis [12,13,14]. In the treatment of knee arthritis, applicators are placed in the meniscus and subchondral bone tissue of the tibia [15]. For the treatment of osteoarthritis, shockwave loading with different intensity loading may be effective [16].

Despite the available results, local mechanical changes in the tissues due to acoustic (mechanical) loading of various intensities remain insufficiently understood because of the limitations of experimental studies. The use of computational (in-silico) methods can significantly help in elucidating the mechanical foundations of bone and cartilage tissue regeneration under conditions of low-energy exposure.

The meniscus is the damping area between the femur and the tibia. Due to its special mechanical characteristics, forces are redistributed, and the load on the proximal tibia is reduced [17]. In osteoarthritis, the meniscus is one of the first to change its mechanical characteristics and is also subject to destruction [18]. Therefore, much attention is paid to the study of the conditions for the regeneration of the tissues of the meniscus. To describe the mechanical behavior of meniscus tissues in the nineties of the last century, single-phase viscoelastic models were used [19]. However, such models do not take into account the emerging resistance force from the fluid flow through the pores of the tissues, which, in turn, affects the incorrectness in the description of the mechanical behavior of the tissues under dynamic loading [20,21,22]. In addition, the assumption about the two-parameter condition for the regeneration of biological tissues implies taking into account the liquid. Currently, for studying the mechanical behavior of the knee joint under dynamic action, two-phase poroelastic models are used to describe the material of the meniscus [23]. A cartilage plate performs the antifriction function and promotes energy dissipation in the knee joint [24,25]. It is known that the cartilage tissue of the knee joint has low permeability; this property of the tissue prevents the rapid outflow of the fluid under dynamic loading [26,27]. Therefore, for the numerical study of the conditions of regeneration under dynamic action, two-phase poroelastic models are preferred [28]. Bone tissues may be described by one-phase and two-phase poroelastic models. Poroelastic models are mainly used for the numerical study of the conditions of bone tissue regeneration [29].

Regarding the numerical models devoted to the simulation of the mechanical behavior of the knee joint in the conditions of shockwave therapy, it should be noted that there is no published study that considers the knee as a joint of two bones with articular cartilages and meniscus: there are only the abovementioned publications, which consider different tissues of the joint.

The aim of this work is to develop a numerical model for shock wave exposure on the knee joint as a whole. As a modeling method, the method of movable cellular automata was adopted, which is a representative of computational particle mechanics, and makes it possible to explicitly describe the generation and development of damage in heterogeneous materials. To achieve this goal, a 3D numerical model of the knee joint was verified and validated. Numerical studies were carried out on external shock wave exposure on the knee joint with different energy flux densities.

## 2. Materials and Methods

### 2.1. Method of Movable Cellular Automata

To describe the mechanical behavior of biological tissues, herein we used the model of a poroelastic body, implemented in the method of movable cellular automata (MCA) [30,31], which is an efficient method of computational particle (discrete) mechanics. It has been established that discrete methods have proven themselves to be very promising for modeling contact loading of different materials at both the macroscale and the mesoscale [32,33]. In the MCA method, a solid is considered as an ensemble of discrete elements of finite size (cellular automata) that interact with each other according to certain rules, which, within the particle approach, and due to many body interaction forces, describe the deformation behavior of the material as an isotropic elastoplastic body. The motion of the ensemble of elements is governed by the Newton–Euler equations for their translation and rotation. Within the framework of the method of movable cellular automata, the value of averaged stress tensor in the volume of an automaton is calculated as a superposition of forces that act to the areas of interaction of the automaton with its neighbors [33]. It is assumed that stresses are homogeneously distributed in the automaton volume. Knowing the components of the averaged stress tensor allows adapting different models of plasticity and fractures of classical mechanics of solids to MCA.

The description of the fluid-saturated material in the MCA method is based on the use of effective (implicit) characteristics, such as the volume fraction of interstitial fluid, porosity, permeability, and the ratio of the macroscopic bulk modulus of elasticity to the bulk modulus of the solid skeleton of the material [34]. The fluid filtration in the material is governed by Darcy’s law. The mechanical effect of pore fluid on stress and strain of the solid skeleton of the automaton is described using Biot’s linear poroelasticity model; therefore, pore fluid pressure affects only the diagonal components of the stress tensor [35].

Details of the method itself and the scheme for numerical solution of the governed equations are given in Appendix A. A three-dimensional version of the method has been implemented in the in-house code MCA3D, which is written in C++ programming language and utilizes Qt library for the pre- and post-processing stages. This code was used in many papers of the authors and their colleagues, in particular, the verification and validation of poroelastic models of tissues of the femur and tibia, based on the MCA method, were carried out in [36,37,38]. A free code for the MCA method solver is also available as a part of the LIGGGHTS package [39]; however, it now provides a reduced functionality for elastic–plastic materials only.

### 2.2. Model of the Knee Joint

The geometric model of the knee joint constructed in this work included the following elements: the epiphyseal part of the femur, the proximal part of the tibia, the cartilage plates, and the meniscus. Each bone consists of a cortical shell and inner cancellous part (Figure 1).

The knee joint was placed in a box imitating an articular (synovial) capsule, consisting of the interior fibrous tissue and an outer shell (Figure 2a). The applicator was modeled as a thin copper plate of a square shape with a size of 20 × 20 × 1 mm^3^ and was located in the meniscus region (black square in Figure 2c). In order to simplify the model and analysis of the results obtained, as well as due to the absence of the need to observe the processes taking place inside the patella, we neglected the presence of the patella.

Standard CAD models available on the Internet (https://www.3dcadbrowser.com/3d-model/human-knee-joint, accessed date 1 December 2021) were used as the component parts of the knee joint.

The poroelastic properties of the biological tissues of the knee joint adopted in this model are presented in Table 1 and correspond to the data in [38,39]. The fluid in bone tissues is assumed to be equivalent to salt water, with the bulk modulus K_f_ = 2.4 GPa, and the density ρ_f_ = 1000 kg/m^3^ [40,41].

In this model, the loading mimicking shockwave exposure was applied using the copper plate as an applicator. To describe the material of the plate, a model of an elastic body was used with the following parameters: density ρ = 8950 kg/m^3^, bulk modulus K = 115 GPa, shear modulus G = 41.6 GPa [42].

For simulation, we used a computer with Intel i9-10980XE CPU and 64 Gb RAM, running the CentOS 8 operating system. The code MCA3D, implementing the MCA method, utilized parallel computing based on the OpenMP library, so typical computation on 36 threads took about 30–50 h, depending on the number of elements.

## 3. Results

### 3.1. Model Verification

The main purpose of verification is to assess the correctness and efficiency of the numerical scheme for solving the governing equations of the method. The key component of numerical model verification is the analysis of the convergence of the obtained results with increasing the resolution of the discrete model (decreasing the discrete element size in the case of computational particle mechanics). The discrete representation of the model is considered optimal when a further increase in its resolution gives no more than a 5% difference with the available resolution [43].

In this work, the analysis for the convergence of a three-dimensional model of the knee joint was carried out by studying the stiffness of the system and the pattern of equivalent strain distribution at different discretization of the considered geometric model (Figure 2) under its uniaxial compression. Herein, the size (diameter) of discrete elements (automata) in the model sample varied from 0.75 mm to 2.0 mm. The compression of the model samples along the vertical direction was carried out by setting a constant velocity of 0.001 m/s to the upper layer of the particles (Figure 2b).

The results on the convergence of the stiffness of the model knee joint showed that the convergence is nonlinear, and the total scatter between the values for the minimum and maximum size of elements does not exceed 2% (Figure 3). A very small difference between the values for the size of elements smaller than 1.3 mm indicates a good accuracy of the numerical model for determining its integral parameters.

However, the pattern of distortional strain distribution (Figure 4) indicates sufficient accuracy in determining the zones of strain concentration only for the models with the size of elements of 0.75 and 1.0 mm. Since the calculation times for these models are 13 h and 6 h, respectively, we will take a sample with the diameter of automates equals to 1 mm as the optimal for subsequent calculations.

### 3.2. Model Validation

The validation of the used poroelastic models of materials for the bone tissues of the knee joint was performed and published in the authors’ previous papers [36,37,44].

Validation of the total model of the entire knee joint was carried out by comparing the simulation results with experimental data from [45], and with other numerical simulations from [46]. For this purpose, a compressive load was applied to the upper layer of the femur by its displacement by 0.3 mm for 1 s in accordance with the reference experiment (Figure 2b).

As mentioned above, the tissues of the cartilaginous plates are subject to the greatest degenerative changes; therefore, the model was validated by comparing the distribution of contact pressure (equivalent stresses in the contact zone) and fluid pressure in these tissues. The distributions of fluid pressure in the pores obtained from our calculations (Figure 5a) are in good qualitative and quantitative agreement with the data presented in the literature [46].

Comparison of the plots for force versus displacement (Figure 5b), which characterizes the rigidity of the system, also showed good agreement with the experimental data presented in [45].

### 3.3. Simulation of Shockwave Exposure on Knee Joint

The main characteristic of shockwave loading is the value of the energy flux density (*PII*), which, according to [47], can be expressed through the product of the acoustic wave intensity (*I*) and normalized time of positive pressure (*T_p_*), as follows:(1)$$PII\equiv I\cdot T_{p}$$
where *T_p_* can be defined as the time to reach 90% of maximum positive pressure (Figure 6). At the same time, the intensity (*I*) is a characteristic of the acoustic impedance of the medium and, in accordance with Equation (1), we obtain an expression for calculating the energy flux density, as follows:(2)$$PII=\frac{v^{2}\rho cT_{p}}{2}$$

The parameter of “normalized pulse length” was determined from the graphs of pressure versus the time of calculation.

In accordance with the mechanobiological principles, the patterns of the distribution of hydrostatic pressure, the distortional (von Mises) strain, and the fluid pressure in the pores were analyzed.

When analyzing the pattern of hydrostatic pressure distribution, it was found that at loading with energy flux density of 0.12 mJ/mm^2^ in the area of cartilaginous plates, the conditions for the onset of the processes of osteogenesis and chondrogenesis (hydrostatic pressure higher than 3 kPa) were achieved (Figure 7a). The pattern of distribution of fluid pressure in the pores (Figure 8a) indicates the fulfillment of conditions (the values of this parameter must lie in the range from 40 kPa to 2 MPa) for the transfer of biological cells. However, the pattern of distortional strain (Figure 9a) indicates the minimum level of values of promoting cartilaginous tissue regeneration in small local areas in the vicinity of the loading plate. These results show the insufficient level of the power of shockwave loading for regeneration on wide areas.

Under shockwave loading with an energy flux density of more than 0.3 mJ/mm^2^ in the regions of cartilaginous plates, the conditions formed for the regeneration of cartilaginous tissues [7,8,9,10]: the value of compressive stresses exceeds 0.15 MPa (Figure 7b,c), and the values of distortional strain lie in the range from 0.05% to 1% (Figure 8b,c) at values of fluid pressure higher than 20 kPa (Figure 9b,c).

With an increase in the intensity of ultrasonic exposure, the amplitude of the hydrostatic pressure and the area of exposure increase, but the patterns remain practically the same (Figure 7b,c, Figure 8b,c and Figure 9b,c).

Compressive stresses with maximum amplitude (0.2–1.5 MPa) are concentrated in the region of the cartilaginous plates and meniscus near the loaded surface. With an increase in the amplitude of the energy flux density, the amplitude of the hydrostatic pressure increases, but the pattern remains the same. In the region of maximum stresses, there is also a maximum of values of shear strain up to 0.6% and fluid pressure up to 1 MPa. At the same time, in the bone tissues adjacent to the cartilaginous plate, stresses up to 0.2 MPa are predominantly observed, the shear strain is close to zero, and the fluid pressure in the pores is about 25–30 kPa. This pattern of the distribution of pressure and deformation indicates the creation of conditions for the regeneration of cartilaginous and meniscus tissues, as well as bone tissues near the loaded surface.

## 4. Discussion

Shockwave loading of various intensities is used for the regeneration of biological tissues. The acoustic effect is aimed at the fusion of bone tissues (fractures) and the regeneration of tissues subjected to degeneration (cartilage, meniscus). Different types of treatment require a different area of application and power of exposure. It is also necessary to take into account the general degenerative changes in the surrounding bone tissues. There are conflicting data on the effective dosage of shockwave therapy. In addition, ethical restrictions do not allow us to accurately establish the patterns of mechanical stimulation of biological tissues.

In this work, by means of numerical modeling, shockwave loading of various intensities on the knee joint with healthy tissues in the region of the cartilaginous plate of the tibia and meniscus was reproduced. The simulation results indicate the localization of the shock wave action in the area of the loading plate. With an increase in the amplitude of the exposure, the volume in which conditions for tissue regeneration are fulfilled increases. Conditions for chondrogenesis are formed in the cartilage plates, which in turn is consistent with the experimental data on shockwave treatment of the knee joint [15,48,49,50]. In addition, conditions for chondrogenesis are observed in the tissues of the meniscus, which is also consistent with experimental data [51].

Different authors report conflicting data on the recommended intensity of shockwave loading for the regeneration of knee tissue [12,13]. Earlier, numerical studies on acoustic exposure effects have been performed mainly on samples of elementary geometry of biological tissues and biomaterials [52,53]. However, such modeling does not take into account geometric and other physiological features (including the presence of intraosseous fluid) [54] which significantly influence the pressure distribution pattern.

The results of our numerical studies have shown that under shock-wave loading above 0.3 mJ/mm^2^, conditions for the regeneration of the tissues of the knee joint could be achieved. The amplitude of the exposure significantly depends on the volume in which the conditions for tissue regeneration are fulfilled.

## 5. Conclusions

The presented numerical model was used for studying shockwave exposure of varying intensity on the knee joint. Analysis of the distribution of hydrostatic pressure, distortional strain, and fluid pressure showed that under shockwave exposure with an energy flux density below 0.3 mJ/mm^2^, conditions for the regeneration of cartilaginous tissues are not formed in the model sample. Under loading with energy flux density higher than 0.3 mJ/mm^2^, the amplitudes of the compressive stress, fluid pressure, and distortional strain corresponding to the conditions of regeneration of cartilaginous tissues are observed in the model samples. Thus, the results obtained are in good agreement with experimental data available from the literature.

In summary, we can conclude that the numerical model of the knee joint developed in this study allows performing a correct simulation of the mechanical behavior of this biological system under shockwave treatment.

## Figures and Tables

**Figure 1 materials-14-07678-f001:**
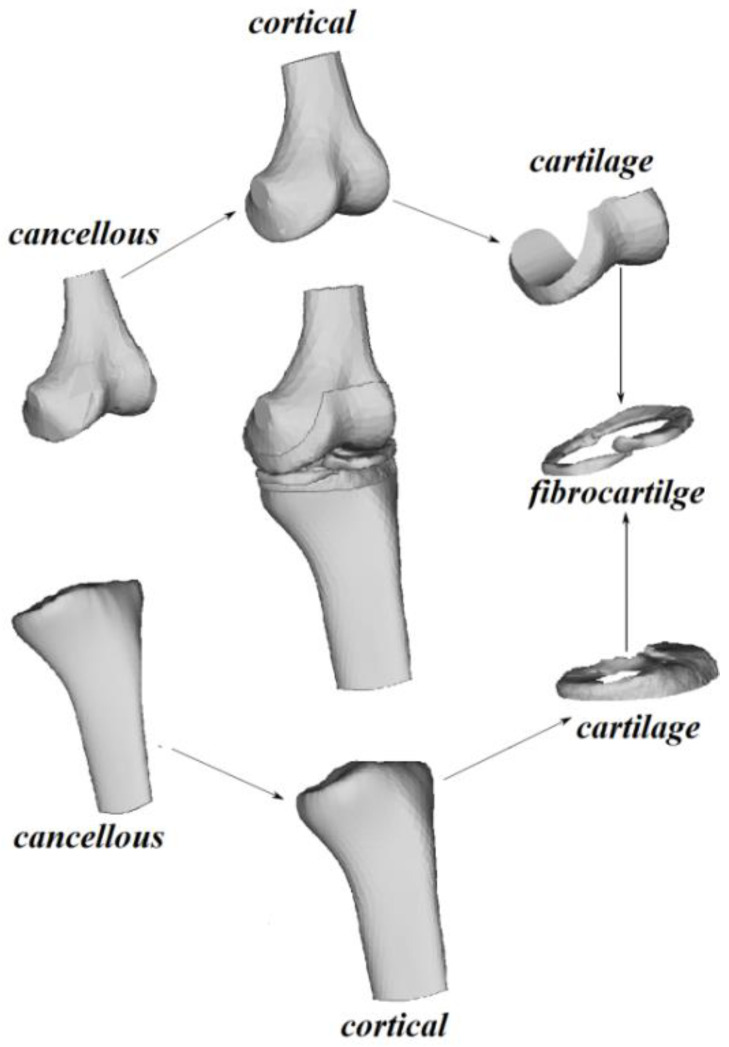
The CAD model of the knee joint and its component parts.

**Figure 2 materials-14-07678-f002:**
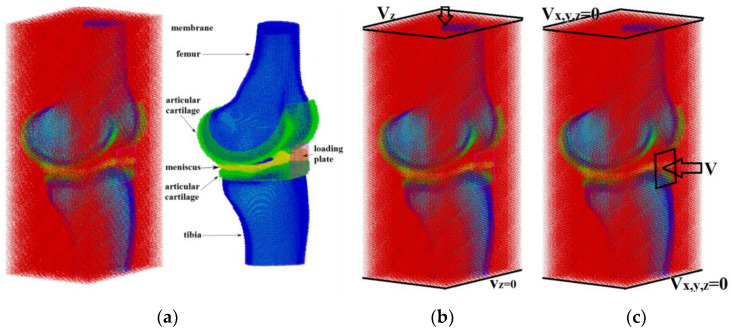
The model of the knee joint: (**a**) structure model of the knee joint; (**b**) loading conditions used for verification and validation; (**c**) loading conditions for simulation of the shock wave exposure.

**Figure 3 materials-14-07678-f003:**
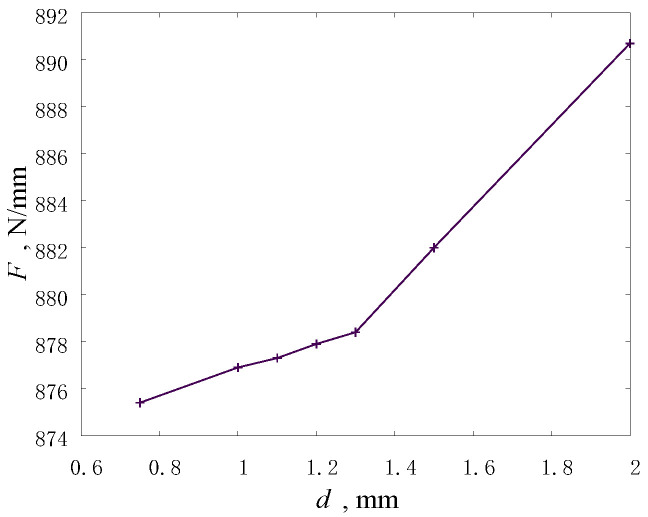
Stiffness of the model knee joint versus the size of discrete elements in the models.

**Figure 4 materials-14-07678-f004:**
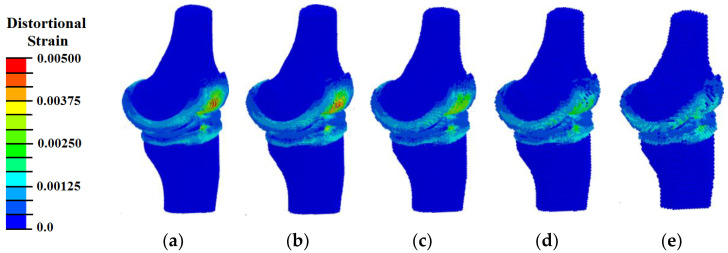
Distributions of distortional strain in the samples with different diameter of elements: (**a**) 0.75 mm; (**b**) 1.0 mm; (**c**) 1.3 mm; (**d**) 1.5 mm; (**e**) 2.0 mm.

**Figure 5 materials-14-07678-f005:**
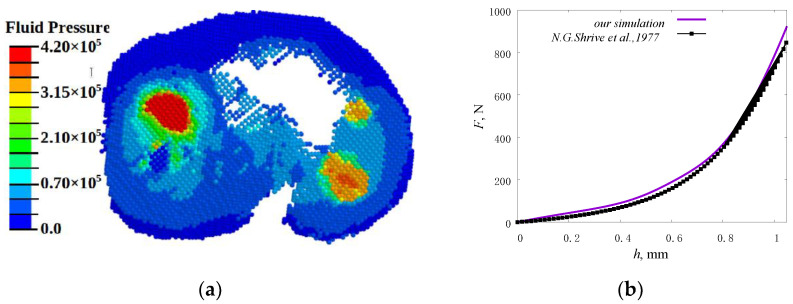
Results of the model validation: (**a**) distributions of fluid pressure (Pa) in the tibia cartilage pores (our simulation); (**b**) plots for applied force versus displacement.

**Figure 6 materials-14-07678-f006:**
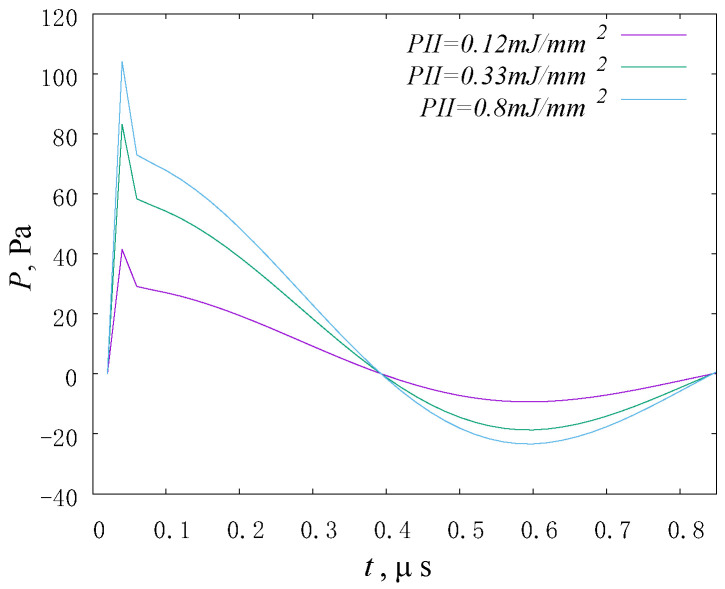
Plots of the pressure profile of shock wave with a different energy flux density.

**Figure 7 materials-14-07678-f007:**
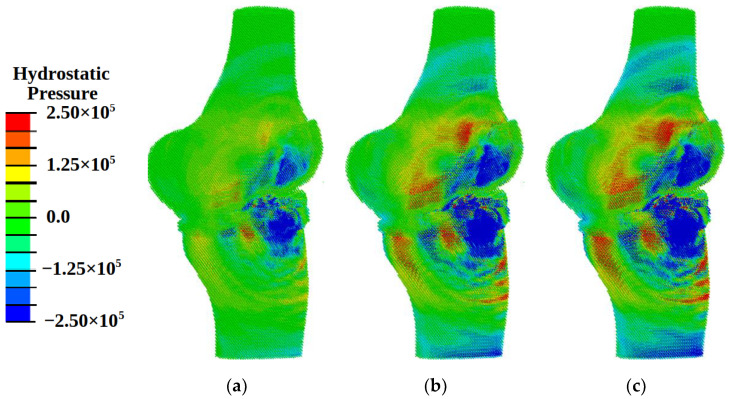
Fields of hydrostatic pressure (Pa) at different energy flux density of shock wave: (**a**) 0.12 mJ/mm^2^; (**b**) 0.33 mJ/mm^2^; (**c**) 0.8 mJ/mm^2^.

**Figure 8 materials-14-07678-f008:**
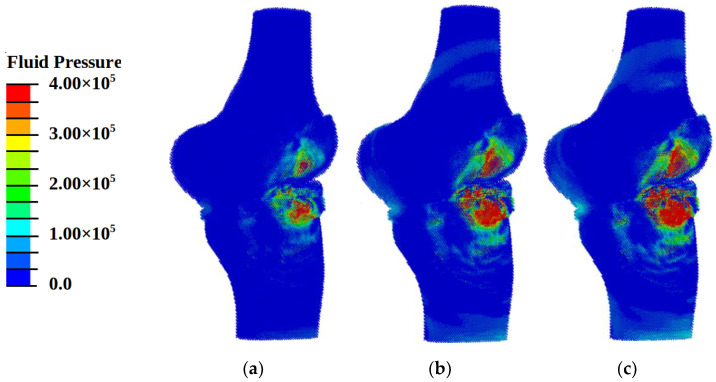
Fields of fluid pressure (Pa) in pores at different energy flux density of shock wave: (**a**) 0.12 mJ/mm^2^; (**b**) 0.33 mJ/mm^2^; (**c**) 0.8 mJ/mm^2^.

**Figure 9 materials-14-07678-f009:**
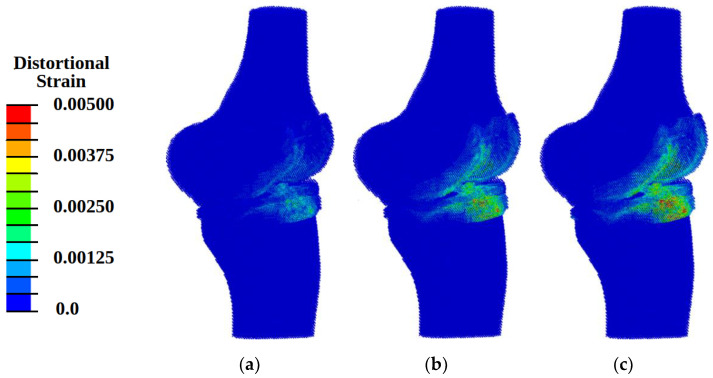
Fields of distortional strain at different energy flux density of shock wave: (**a**) 0.12 mJ/mm^2^; (**b**) 0.33 mJ/mm^2^; (**c**) 0.8 mJ/mm^2^.

**Table 1 materials-14-07678-t001:** Elastic and poroelastic parameters of the bone tissues.

Type of Tissue	Density of the Matrix, ρ, kg/m^3^	Shear Modulus of the Matrix, G, GPa	Bulk Modulus of the Matrix, K, GPa	Bulk Modulus of the Solid, Ks, GPa	Porosity, θ	Permeability, k, m^2^
cortical	1850	5.55	14	17	0.04	3.6 × 10^−15^
cancellous	700	1.3	3.3	15	0.7	1.0 × 10^−11^
cartilage	800	0.0043	0.00416	3.4	0.8	4.8 × 10^−18^
fibrocartilage	900	0.130	0.283	2.3	0.8	9.5 × 10^−19^
fibrous	1000	0.00043	0.00416	2.3	0.9	7.5 × 10^−19^
capsule shell	1000	1.2	2.3	2.3	0.9	1.0 × 10^−11^

## Data Availability

The data that support the findings of this study are available from the corresponding author upon reasonable request.

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
