# Peer review of "Numerical Modeling of Shockwave Treatment of Knee Joint"

_materials, 2021, doi:10.3390/ma14247678_

Round 1

Reviewer 1 Report

Majors:

  1. What is 3D shape or volume of knee joint anatomy and its mathematical description? Please add a part for these in Materials and Methods.
  2. How to make simulations, from what system, what software? What is time consumption? Please provide clear information about these concerns in Materials and Methods.

Minors

  1. Line 12, in Abstract, the last sentence of “The results obtained indicate the creation of …. ” is very unclear. Please reshape this sentence and make a clear conclusion on this study.
  2. Line 140, in Materials and Methods, the sentence of “Standard CAD models available on the Internet … ”, where users can find these models, please provide a valid link for it.
  3. There are two Fig 7, please check and correct them.
  4. From Fig 3 to Fig 7, except Fig 5, number of colour bar indicators is very difficult to read, please give only minimum and maximum only.

Author Response

What is 3D shape or volume of knee joint anatomy and its mathematical description? Please add a part for these in Materials and Methods.

Thank you for this question, we agree that the images shown in Fig.1 were not enough for understanding the geometrical details of our model of such a complex object as the knee joint; a mathematical description was also omitted.

In the revised manuscript, we added the detailed 3D views of all geometrical parts of the model in Fig. 1 as well as the detailed mathematical description of the MCA method in Appendix A.

How to make simulations, from what system, what software? What is time consumption? Please provide clear information about these concerns in Materials and Methods.

Thank you for this question, we agree that it was not clear in the manuscript.

We provided all the necessary information about the MCA method in Appendix A. So, now it becomes clear that our simulation is just a numerical solution of the equations of motion (Eq. A1). The loads mimicking uniaxial compression (for verification and validation), and shockwave loading through copper applicator (for numerical research) are shown in Fig. 1.

The MCA-method has been implemented in our own in-house software MCA3D, which is written in C++ programming language and utilizes Qt library for the pre- processing (creating geometry, loads, setting material properties, etc.) and post-processing (viewing and analyzing the results, saving images of the model and distribution of stresses and strains, etc.) stages of the simulation process. For computation, we used a computer with Intel i9-10980XE CPU and 64 Gb RAM running CentOS 8 operating system. The code MCA3D implementing the MCA method utilized parallel computing based on the OpenMP library, so typical computation on 36 threads took about 30-50 hours, depending on the number of elements. We provided this information in “Materials and Methods” section.

Minors

Line 12, in Abstract, the last sentence of “The results obtained indicate the creation of …. ” is very unclear. Please reshape this sentence and make a clear conclusion on this study

We restated the last sentence of the Abstract in a different way that should make it clear that we found the level of the shockwave load, which is necessary for the regeneration of cartilage tissues and meniscus, i.e. to get therapeutic effect.

Line 140, in Materials and Methods, the sentence of “Standard CAD models available on the Internet … ”, where users can find these models, please provide a valid link for it.

We added the URL of the Internet resource where these models can be downloaded from (https://www.3dcadbrowser.com/3d-model/human-knee-joint) at the end of the first paragraph of section 2.1.

There are two Fig 7, please check and correct them.

Thank you, we corrected this typo.

From Fig 3 to Fig 7, except Fig 5, number of colour bar indicators is very difficult to read, please give only minimum and maximum only.

We made it as you wish. Now the numbers can be clearly seen.

Reviewer 2 Report

(1)“2.1. Method of movable cellular automata”, the writing style of this part is the same as that of the introduction. It is suggested to give key theoretical formulas or frameworks, necessary illustrations, etc.

(2) Supplement the constitutive relationship of materials involved in the model of knee joint.

(3) There are too few data points in Figure 2. It is recommended to supplement data points, especially near d = 1.3mm.

(4) For the distribution of fluid pressure in Fig. 4 (a), it is necessary to give and clearly identify the results of this paper and the results of literature [44].

(5) For the force displacement curve in Fig. 4 (b), when the load is in the range of 100N to 400N, the solution in this paper is quite different from that in the literature, and the curve shape is completely different. It is necessary to analyze the reasons and improve the model.

Author Response

(1)“2.1. Method of movable cellular automata”, the writing style of this part is the same as that of the introduction. It is suggested to give key theoretical formulas or frameworks, necessary illustrations, etc.

We added Appendix A with the detailed description of the method of movable cellular automata.

(2) Supplement the constitutive relationship of materials involved in the model of knee joint.

We added Appendix A, where also provided the detailed description of poroelastic model used for simulation.

(3) There are too few data points in Figure 2. It is recommended to supplement data points, especially near d = 1.3mm.

Yes. We added data points for d = 1.1 mm and d = 1.2 mm in Fig. 2 (see p.5).

(4) For the distribution of fluid pressure in Fig. 4 (a), it is necessary to give and clearly identify the results of this paper and the results of literature [44].

Yes. We added information about the location of the pattern to be compared in [46]. The pattern from [46] was not presented in this work due to the requirement to have permission for its publication.

Distributions of fluid pressure (Pa)
in the tibia cartilage pores (our simulation)

Fluid pressure (MPa) in the tibia cartilages
(Fig. 3a on page 284 in [46])

(5) For the force displacement curve in Fig. 4 (b), when the load is in the range of 100N to 400N, the solution in this paper is quite different from that in the literature, and the curve shape is completely different. It is necessary to analyze the reasons and improve the model.

We agree. We made additional computations and changed the plot. Bad agreement of the origin curve in the range from 100 to 400 N is connected with an incorrect loading conditions (the loading velocity was being increased from zero to the maximum value very quick, i.e. “acceleration time” was too short) and, as a consequence, with the emerging dynamic effect. Below is a picture of the influence of the “acceleration time” on the curve for force versus displacement.

Plots of force versus displacement for different “acceleration time”
(T1=2.5 µs, T2=10.0 µs, T3=20.0 µs)

Round 2

Reviewer 2 Report

Fig. 6 can also be improved, the values of different speed of loading can be directly marked in the figure, like the style of Fig. 5 (b).

Author Response

Fig. 6 can also be improved, the values of different speed of loading can be directly marked in the figure, like the style of Fig. 5 (b).

We changed the picture in Fig. 6 to be in the same style as Fig. 5 (b), and proofread the text to improve the English.

Thank you for your help in improving the quality of our manuscript.